# A Novel Curcumin Arginine Salt: A Solution for Poor Solubility and Potential Anticancer Activities

**DOI:** 10.3390/molecules28010262

**Published:** 2022-12-28

**Authors:** Adel Al Fatease, Mai E. Shoman, Mohammed A. S. Abourehab, Heba A. Abou-Taleb, Hamdy Abdelkader

**Affiliations:** 1Department of Pharmaceutics, College of Pharmacy, King Khalid University, Abha 62529, Saudi Arabia; 2Department of Medicinal Chemistry, Faculty of Pharmacy, Minia University, Minia 61519, Egypt; 3Department of Pharmaceutics, College of Pharmacy, Umm Al-Qura University, Makkah 21955, Saudi Arabia; 4Department of Pharmaceutics, Faculty of Pharmacy, Minia University, Minia 61519, Egypt; 5Department of Pharmaceutics and Industrial Pharmacy, Faculty of Pharmacy, Merit University (MUE), Sohag 82755, Egypt

**Keywords:** curcumin, L-arginine, cytotoxicity, breast cancer

## Abstract

Curcumin is a natural polyphenolic compound with well-known anticancer properties. Poor solubility and permeability hamper its use as an anticancer pharmaceutical product. In this study, L-arginine, a basic amino acid and a small hydrophilic molecule, was utilized to form a salt with the weak acid curcumin to enhance its solubility and potentiate the anticancer activities of curcumin. Two methods were adopted for the preparation of curcumin: L-arginine salt, namely, physical mixing and coprecipitation. The ion pair or salt was characterized for docking, solubility, DSC, FTIR, XRD, in vitro dissolution, and anticancer activities using MCF7 cell lines. The molecular docking suggested a salt/ion-pair complex between curcumin and L-arginine. Curcumin solubility was increased 335- and 440-fold by curcumin in L-arginine, physical, and co-precipitated mixtures, respectively. Thermal and spectral analyses supported the molecular docking and formation of a salt/ion pair between curcumin and L-arginine. The cytotoxicity of curcumin L-arginine salt significantly improved (*p* < 0.05) by 1.4-fold, as evidenced by the calculated IC_50%_, which was comparable to Taxol (the standard anticancer drug but with common side effects).

## 1. Introduction

Curcumin has been one of the most investigated phytochemical substances for decades. Curcumin is a yellow spice or pigment with a pungent taste extracted from turmeric. The latter is cultivated in many countries, including India, China, Brazil, Peru, and Sri Lanka [1,2]. The increasing interest in curcumin is due to the potential health benefits claimed for it in treating several chronic diseases, such as different types of cancer (breast, colon, and prostate cancers), cardiovascular diseases, liver diseases, and neurological diseases (Alzheimer’s and Parkinson’s diseases) [3].

The role of curcumin has been recently highlighted in the prevention of many types of cancer due to its high safety, low side effects, and availability at a low cost [4]. Therefore, curcumin has been included in many foods and nutraceutical products. However, due to several challenges, there is still a long way for curcumin to be developed as a pharmaceutical product [2]. Poor solubility, permeability, bioavailability, and extensive degradability are the most common challenges for developing curcumin as one of the most promising pharmaceuticals [4].

Curcumin is a crystalline powder with an orange color. It is a highly hydrophobic substance with poor aqueous solubility. Moreover, it is a polyphenolic compound, and its solubility increases in alkaline conditions due to the deprotonation and ionization of phenolic functional groups in curcumin [2,5]. Furthermore, it has three weak acidic groups (hydroxyl and two phenolics) with pKa values of 8.38, 9.88, and 10.51 [2].

It has high stability in acidic conditions but minimal solubility. In addition, it is fully unionized and has very low solubility at acidic and neutral (pH 2–7) conditions. Therefore, the pH of the physiological fluid along the gastrointestinal tract discourages curcumin’s solubilization, which can explain the poor absorption and bioavailability of oral curcumin [6]. This requires large doses to obtain considerable oral absorption, resulting in increased gastrointestinal side effects, such as gastric irritation, nausea, and diarrhea [7]. Nevertheless, curcumin has been reported to be chemically stable in the stomach and intestine for a sufficient time to allow its absorption if solubilized [6]. Therefore, the formation of an ion pair or salt of the weak phenolic compound with the basic amino acid would be an innovative idea to be investigated. The solubility of curcumin in the stomach is low due to poor ionization in the gastric acid; thus, enhancing its solubility in the gastric fluid might improve the overall oral absorption and bioavailability of curcumin by increasing the pH of the stagnant diffusion layer surrounding drug particles using alkalinizing excipients, such as L-arginine. This technique successfully formulated dispersible oral aspirin tablets using sodium bicarbonate and weakly acidic drugs, such as aspirin. However, excessive use of sodium bicarbonate might increase sodium load and blood pressure [8]. Therefore, alkalinization by sodium bicarbonate is not a suitable solution for treating chronic diseases in elderly patients. The use of basic amino acids, such as lysine (pKa = 10.8) and arginine (pKa = 12.5) has been successfully investigated for enhancement of solubility and permeability of weak acid (e.g., indomethacin) and natural polyphenolic drugs (e.g., quercetin and rutin) [9,10]. Amino acids are natural biomolecules and are safe and nontoxic substances. Furthermore, the basic amino acids, such as lysine and arginine, have been reported to improve cell permeability and cellular uptake of some hydrophilic macromolecules (e.g., insulin) and weak acid small drugs (e.g., indomethacin) by forming ion-pair complexes capable of penetrating cells and triggering amino acid transporters to enter the cells [9,11]. The current study aimed to investigate the possibility of preparation and characterization of curcumin–arginine ion pair/salt to improve curcumin solubility in the first part of the gastrointestinal tract under acidic gastric conditions. Docking studies and spectral and thermal analyses were employed to characterize the prepared curcumin arginine salts using two different physical and coprecipitated mixing methods. L-arginine was selected for being a stronger basic amino (pKa = 12.5) than L-lysine (pKa = 10.8); therefore, it is likely to form a more stable ion pair/salt with the weak acid drug curcumin.

## 2. Results and Discussion

Amino acids are natural compounds considered regulatory safe and nontoxic excipients. Over the past decade, using amino acids as water-soluble excipients to enhance poorly soluble drugs has gained growing interest [9,12,13]. Depending on the drug ionization state, the mechanism of improving drug solubility varies. Weak acid and basic drugs can form salts with basic (e.g., L-arginine and L-lysine) and acidic amino acids (e.g., aspartic acid), respectively, while nonionizable medications (e.g., carbamazepine) and amino acids (neutral, acidic, and basic members) can work as hydrotropes and form ion pairs and hydrophobic attractions with the nonionizable drugs [12].

For example, both water solubility and permeability of the bioflavonoid quercetin were significantly enhanced by forming an ion pair complex with the basic amino acid lysine [13]. More interestingly, L-arginine showed better solubility-enhancing effects of alkyl gallate than lysine, suggesting stronger interactions of L-arginine with the poorly soluble drug alkyl gallate [14].

Curcumin is a polyphenolic compound with two phenolic functional groups and one enol group that imparts curcumin’s weak acidic characteristics. Curcumin is a hydrophobic drug with log *p* = 3 and can be classified as a class II drug with poor solubility and high permeability [15]. This study investigated L-arginine’s basic amino acid capability to form an ion pair complex or salt with curcumin to enhance curcumin’s solubility and permeability. The curcumin–L-arginine mixture was produced through two methods: (1) the physical mixing of the two powders together in a small evaporating dish with a spatula; and (2) the solvent evaporation method by dissolving curcumin in methanol, mixing it with aqueous solution of L-arginine, and then stirring at a mildly elevated temperature until complete evaporation of the solvent.

### 2.1. Docking

The poses, orientations, and possible interactions of curcumin and L-arginine are shown in Figure 1. The molecular docking study demonstrated that L-arginine was sandwiched between the two phenolic groups and stabilized by electrostatic interaction between the basic amino group of L-arginine and the enol group in the middle of the curcumin structure, with an estimated energy score of −3.515 kcal/mol. While there are scarce reports about visualizing the docking of amino acids with small molecules such as curcumin, the literature supports the electrostatic interactions and cocrystal formation of amino acids with many drugs, such as the polyphenol rutin, mesalamine, itraconazole, and naproxen [16]. More specifically, L-arginine has been reported to increase the solubility of coumarin by forming favorable interactions with the aromatic ring structure [17].

### 2.2. Solubility

The solubilities of curcumin alone and in curcumin-L-arginine PM and Coppt mixtures are shown in Figure 2A. The pH of the tested solutions at equilibrium (started solubility) was measured, and the results are shown in Figure 2B.

The results showed that the solubility of curcumin in water was 2 µg/mL. The solubility increased dramatically to 670 and 880 µg/mL from curcumin-L-arginine PM and Coppt, respectively. These 335- and 440-fold increases in curcumin solubility could be attributed to the increased ionization of the weakly acidic drug curcumin in the presence of the basic amino acid curcumin. The pH of the curcumin–L-arginine PM and Coppt at equilibrium was measured, and they were found to be 8.8 and 8.7, compared to that of curcumin, which was 4.8. Intriguingly, it seemed that the solvent evaporation (coprecipitation) method slightly contributed to the enhancement of curcumin solubility due to possible micronization and better drug wettability from more intimate contacts and interaction at molecular levels during coprecipitation, compared to the physical mixing method.

L-arginine has been reported to show superior solubility-enhancing effects of poorly soluble alkyl gallates, suggesting favorable interactions of L-arginine with the aromatic rings of alkyl gallates (model drug substances) [14].

### 2.3. DSC, FTIR and XRD Spectra

The DSC thermal traces recorded for curcumin, L-arginine, and curcumin–L-arginine physical and coprecipitated mixtures are shown in Figure 3. A sharp endothermic peak was recorded for curcumin at 183 °C. This thermal event was due to the melting of curcumin [18,19]. Two thermal events were reported for L-arginine at 109 °C and 252 °C due to the loss of bound water and arginine melting, respectively [20].

Curcumin–L-arginine PM was a simple superimposition of curcumin and L-arginine. In contrast, curcumin–L-arginine demonstrated the complete disappearance of characteristic thermal events for both curcumin and L-arginine and the appearance of a new thermal event at 178 °C. These results indicated that forming a new complex or salt and the coprecipitation technique were essential for generating this complex or salt.

Figure 4 shows the FTIR spectra of curcumin, L-arginine, curcumin, L-arginine PM, and Coppt mixtures. Curcumin’s FTIR spectrum revealed characteristic bands at 3515 cm^−1^ for phenolic group stretching [21] and at 1610 cm^−1^ for C=O and aromatic C=C stretching vibrations [22]. While these characteristic FTIR bands of curcumin are simple addition and superimposition for curcumin and L-arginine PM, these bands disappeared in the spectra for curcumin and L-arginine Coppt, indicating electrostatic attractions and hydrogen bonding. 

Figure 5 shows XRD diffractograms of curcumin, L-arginine, curcumin, L-arginine PM, and Coppt mixtures. Curcumin demonstrated sharp diffraction peaks, indicating the crystallinity of the drug. Similarly, L-arginine showed strong diffraction peaks. The XRD spectra for curcumin and L-arginine PM were obtained by simple addition and superimposition of the two XRD diffractograms. In contrast, the complete disappearance of the characteristic diffraction peaks for curcumin-L-arginine Coppt indicated electrostatic attraction and hydrogen bonding formation.

### 2.4. In Vitro Dissolution of Curcumin

In vitro dissolution was studied in a challenging dissolution medium where curcumin exhibited very poor solubility using a simulated gastric fluid with pH 1.2 and sodium lauryl sulfate (1% *w*/*w*) to maintain sink conditions (Figure 6).

Slow drug dissolution of up to 30% over 2 h was recorded for curcumin powder. Curcumin–L-arginine PM showed a slight improvement in the dissolution of curcumin; up to 40% of curcumin was dissolved in 2 h. On the other hand, a marked enhancement of curcumin dissolution was recorded in up to 60% of drug releases. This two-fold enhancement of dissolution rates recorded for curcumin and L-arginine Coppt could be attributed to the salt formation and alkalinization of the diffusion layer surrounding curcumin particles and the increased ionization of curcumin and its solubility in the stagnant diffusion layers [13].

### 2.5. Cytotoxicity Studies

Six consecutive concentrations (3.125, 6.25, 12.5, 25, 50, and 100 µg/mL) of curcumin alone and curcumin, L-arginine, Coppt, and taxol were investigated for cytotoxicity on MCF-7 human breast epithelial cell lines. The results of the toxicity study on MCF-7 human breast epithelial cell lines are shown in Figure 7. Marked cell death (70–100%) was recorded in concentrations ranging from 25 to 100 µg/mL for all tested samples (Figure 7). However, the other lower concentrations (6.250 and 3.125 µg/mL) showed a significant reduction in cytotoxicity percentage for curcumin, L-arginine, and Coppt compared to curcumin alone. This was evident from the estimated IC_50%_ for the standard anticancer drug taxol, curcumin and L-lysine salt, and curcumin alone (4.85 ± 0.05, 5.64 ± 0.08, and 7.83 ± 0.03 µg/mL, respectively). These findings can be explained by the fact that L-arginine increased curcumin’s permeability and cell uptake by forming an ion-pair complex. Additionally, other research has suggested that curcumin and L-arginine salts can be transported on surface-decorated L-arginine receptors in cancer cells [9,11]. For example, the basic amino acid lysine enhanced the solubility of insulin in water; further, the permeability of insulin-L-lysine ion pair complex was significantly enhanced through buccal cell layers [11]. Additionally, curcumin–polyethylene glycols soluble complex has been reported to enhance the anticancer potential against colorectal adenocarcinoma [23].

## 3. Materials and Methods

Curcumin and dimethyl sulfoxide (DMSO) were supplied by Alfa Aesar, Fisher Scientific, and Heysham, Lancashire, UK. L-arginine was purchased from Fluka AG, Buchs, Switzerland. Taxol (paclitaxel, 6 mg/mL) was purchased from Bristol-Myers Squibb, New York, NY, USA. Sodium lauryl sulfate was purchased from Winlab Limited, Maidenhead, Berkshire, UK.

### 3.1. Preparation of Curcumin-L-Arginine Physical and Coprecipitated Mixtures

The curcumin–L-arginine physical mixture (PM) was prepared by separately weighing the equivalent of the molar weight in mg. Curcumin and L-arginine were thoroughly mixed in a porcelain dish for 2 min using a spatula and sieved through a 125 µm sieve.

Curcumin was dissolved in methanol (30 mL) and L-arginine was dissolved in distilled water (10 mL). The methanolic curcumin and the aqueous solutions of the basic amino acid were mixed in a porcelain dish (100 mL capacity). The porcelain dish was placed on a hot plate stirrer, LabTech, Daihan, Korea, adjusted at 80 °C, and left until complete evaporation. The casted powder was pulverized using a pestle and sieved through a 125 µm sieve.

### 3.2. Docking

Molecular docking studies were performed using Molecular Operating Environment (MOE) 2014.09 software (Chemical Computing Group, Montreal, QC, Canada) to predict the poses and orientation of L-arginine on the surface of curcumin. The 3D structure of curcumin was built using the builder interface and the energy was reduced to a root mean square deviation (RMSD) gradient of 0.01 kcal/mol using the QuickPrep tool of the MOE software. Moreover, the 3D structures of L-arginine and curcumin were built using MOE Builder and their energy was minimized. L-arginine was docked into the surface of curcumin using an induced-fit docking protocol with the Triangle Matcher method and dG scoring system for pose ranking. Free energy values were elected and reported after a visual assessment of the resultant docking poses with the highest stability and lowest binding.

### 3.3. Equilibrium Solubility Studies

Excess amounts of curcumin and curcumin–L-arginine physical and coprecipitated mixtures were added to 10 mL of distilled water in small conical flasks separately and placed in a thermostatic shaking water bath (Shel Lab water bath, Sheldon Cornelius, OR, USA) at 37 ± 0.5 °C at a speed of 120 strokes per min. The samples were left for 48 h. Aliquots were withdrawn, filtered, and measured spectrophotometrically at 470 nm using a UV-visible spectrophotometer, Jenway Model 6305, Loughborough, England.

### 3.4. Differential Scanning Calorimetry (DSC), Fourier Transfer Infra-Red Spectroscopy (FTIR), and X-ray Diffraction (XRD)

Proper amounts of curcumin, L-arginine, and curcumin–L-arginine complex/salt were transferred into 40 µL capacity aluminum pans. The pan temperature was increased from 30 to 350 °C at 10 °C per min using a DSC calorimeter (DSC-60; Shimadzu, Kyoto, Japan). Nitrogen was used as a purging gas at a flow rate of 20 mL/min.

FTIR spectrophotometry (FT-IR, Tensor 37, Bruker, Billerica, MA, USA) was used to stack the spectra of curcumin, L-arginine, and curcumin–L-arginine complex/salt. Spectra were collected from KBr disks. The spectra were collected in a range of 400 to 4000 cm^−1^.

The crystallinity of curcumin, L-arginine, and curcumin–L-arginine complex/salt was studied using an XRD diffractometer (Unisantis XMD-300, GmbH, Osnabruck, Germany). The tube operated at 45,000 V and 0.8 mA. The scan range was 5–60° of a 2θ diffraction angle.

### 3.5. In Vitro Dissolution

The dissolution medium was composed of simulated gastric fluid at pH 1.2 (900 mL) containing sodium lauryl sulfate (SLS) at 1% *w*/*w* using a dissolution tester (Pharma Test D-63512, Hainburg, Germany). The media were stirred with USP apparatus 2 at 100 rpm and the temperature was set at 37 °C. Curcumin, PM, and Coppt dispersed mixtures weighing 20 mg or equivalent to 20 mg of curcumin were dispersed onto the dissolution flasks. A sample of 5 mL was withdrawn at specified time intervals and replaced with another 5 mL of the fresh dissolution medium. The samples were analyzed spectrophotometrically, as mentioned above.

### 3.6. Cytotoxicity Assays

Human breast MCF-7 epithelial cells (ATCC # HTB-22, ATCC, Manassas, VA, USA) were used in this study. Six consecutive concentrations (3.125 to 100 µg/mL) of curcumin or its equivalence from curcumin, L-arginine, and Coppt were prepared by dilution using the growth medium. A 96-well plate was inoculated with 0.1 mL/well containing 1 × 10^5^ cells/mL and incubated at 37 °C for 24 h [20]. The medium was removed from the 96-well plates and the formed cell monolayer was washed twice with the medium. Two-fold dilutions of the tested sample were made using an RPMI medium with 2% serum [20]. Aliquots of 100 µL each of the diluted samples were added to 3 consecutive wells. The negative control was the fresh medium, and the positive control was paclitaxel (6 mg/mL), used in 6 different concentrations (300 to 1.71 µg/mL). The plate was incubated at 37 °C and examined. Cells were checked for any signs of toxicity, including partial or complete loss of the monolayer, rounding, shrinkage, or cell granulation. An MTT solution was prepared (5 mg/mL in PBS) (Biobasic, Markham, ON, Canada). A volume of 20 µL MTT solution was added to each well, which was then placed in a thermostatically controlled shaker (ASAL 715 CT, Anzio, Italy) at 150 rpm for 5 min to thoroughly mix the MTT into the media and, finally, incubated at 37 °C, 5% CO_2_ for 4 h. The media were discarded and formazan (MTT metabolic product) was dissolved in 200 µL DMSO by shaking at 150 rpm for 5 min. The optical density was measured by a microplate reader (Mindray MR-96A, Shenzhen, China) at 560 nm and by subtracting the background at 620 nm.

### 3.7. Statistical Analysis

Statistical analysis was conducted using GraphPad Prism software, USA, to test for analysis of variance (ANOVA), with a statistical significance set at *p* < 0.05.

## 4. Conclusions

It is concluded that L-arginine can be a promising excipient to improve both solubility and enhance cellular uptake of curcumin. The anticancer activities of curcumin–L-arginine salt were significantly improved, and the IC_50%_ estimated for the salt was comparable to that for Taxol (paclitaxel) injection. The solubility of curcumin was significantly improved by more than 400-fold. These significant findings support curcumin–L-arginine salt as a potential anticancer agent due to its few side effects and high safety profile.

## Figures and Tables

**Figure 1 molecules-28-00262-f001:**
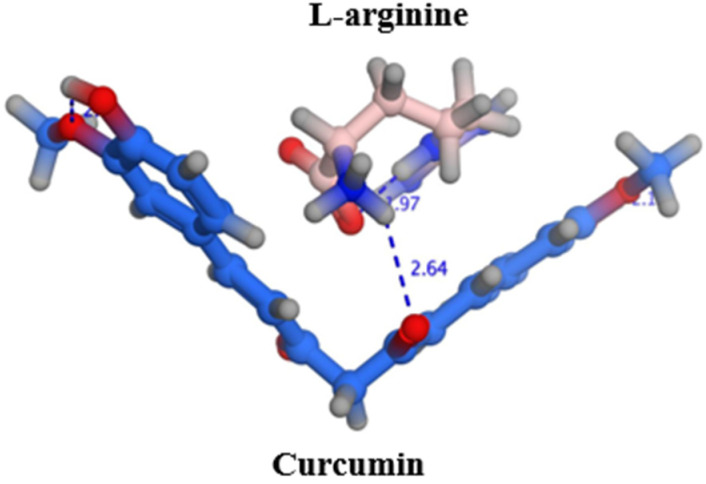
Possible interactions between curcumin and L-arginine. Dashed lines denote H-bond length in Angstrom.

**Figure 2 molecules-28-00262-f002:**
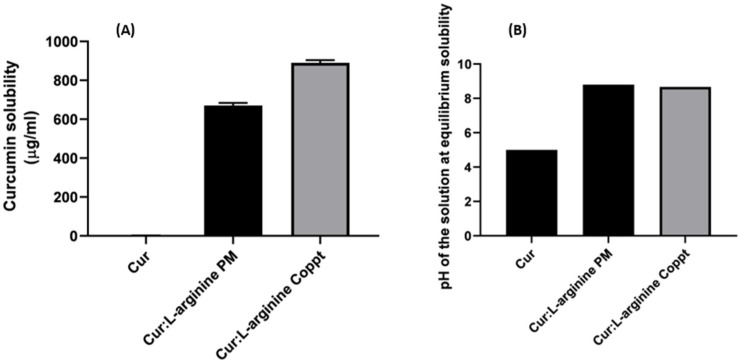
Solubility (**A**) and pH of the solutions measured at equilibrium for curcumin and curcumin–L-arginine physical (PM) and coprecipitated (Coppt) mixtures (**B**) (results are expressed as mean ± SD; *n* = 3).

**Figure 3 molecules-28-00262-f003:**
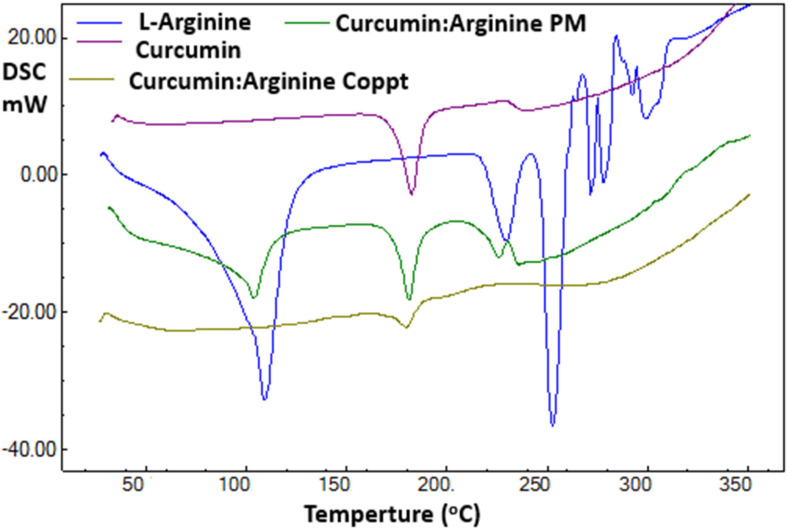
DSC thermal spectra of curcumin, curcumin-L-arginine physical mixture, and coprecipitated dispersion.

**Figure 4 molecules-28-00262-f004:**
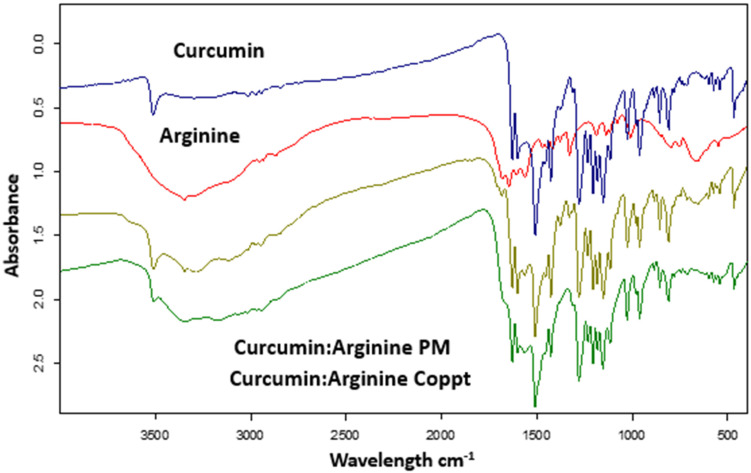
FTIR spectra of curcumin, curcumin-L-arginine physical mixture, and coprecipitated dispersion.

**Figure 5 molecules-28-00262-f005:**
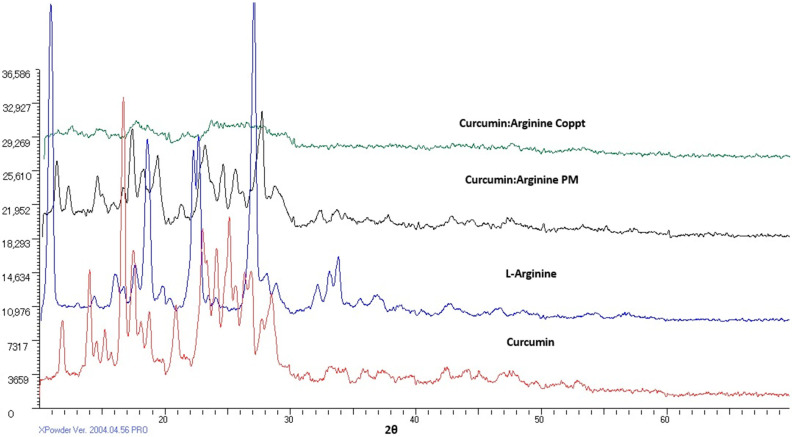
XRD spectra of curcumin, curcumin-L-arginine physical mixture, and coprecipitated dispersion.

**Figure 6 molecules-28-00262-f006:**
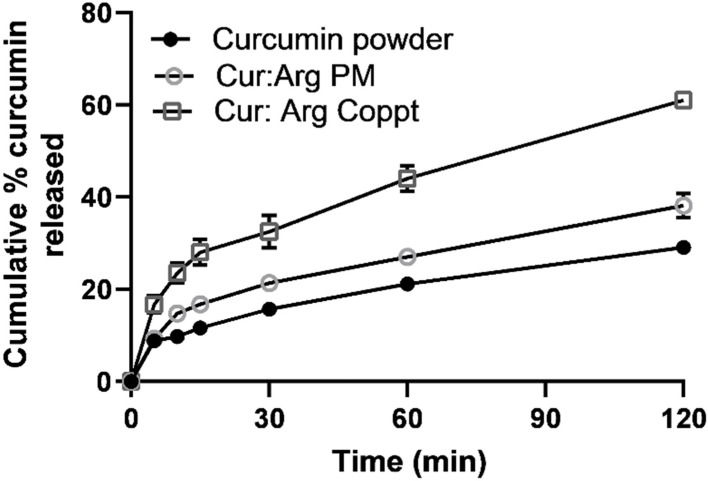
In vitro drug dissolution from curcumin alone, curcumin and L-arginine, and curcumin and L-arginine PM and Coppt (results are expressed as mean ± SD; *n* = 3).

**Figure 7 molecules-28-00262-f007:**
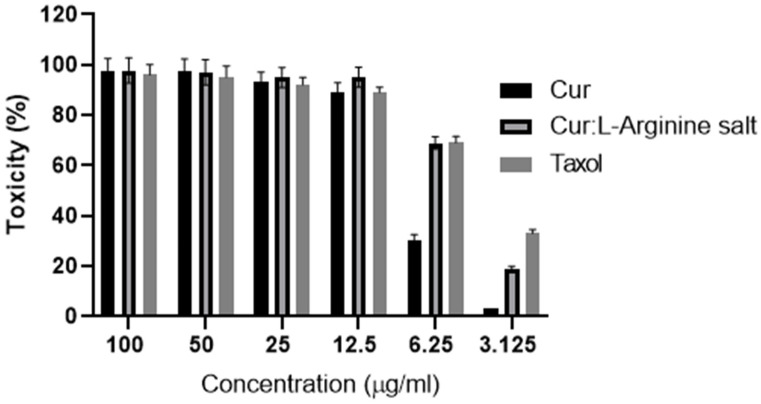
Cytotoxicity (%) of six different concentrations of curcumin, curcumin and L-arginine salt, and paclitaxel (Taxol^®^) using MCF-7 human breast epithelial cell lines (results are expressed as mean ± SD; *n* = 5).

## Data Availability

Upon request.

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
