# Peer review of "A Novel Curcumin Arginine Salt: A Solution for Poor Solubility and Potential Anticancer Activities"

_molecules, 2022, doi:10.3390/molecules28010262_

Round 1
Reviewer 1 Report
23- slat/ion-pair (spelling mistake) complex between curcumin and L-arginine. 335- and
82- In this study, curcumin-arginine complexes/salts were investigated for enhancement of both
solubility and permeability. But the title mentioned solubility and potentiating anticancer
activity- please clarify.
- Why author specifically used L arginine? Why didn’t try with lysine?- please give an explanation.
- Author has studied with Cur, Cur L arginine, Taxol, Bur why author didn’t study with Taxol combined with L arginine? please give an explanation.
Web source/URL of docking software to be quoted as a reference.
- Docking score of the pose selected need to be mentioned.
- Values highlighted in Figure 1 require explanation.
Author Response
We appreciate your constructive comments which are all valuable and very helpful for revising and improving our paper, as well as guiding our research. We have carefully considered comments and have revised the manuscript accordingly. The main revised portions of the manuscript are trackchanged, and our point-by-point responses to the comments are listed below.
23- slat/ion-pair (spelling mistake) complex between curcumin and L-arginine. 335- and
82- In this study, curcumin-arginine complexes/salts were investigated for enhancement of both
solubility and permeability. But the title mentioned solubility and potentiating anticancer
activity- please clarify.
The abstract has now been corrected accordingly.
Why author specifically used L arginine? Why didn’t try with lysine?- please give an explanation.
Both L-lysine and L-arginine are basic amino acids. But L-arginine is stronger base than L-lysine We thought that L-arginine can form more stable salt with L-arginine.
Author has studied with Cur, Cur L arginine, Taxol, Bur why author didn’t study with Taxol combined with L arginine? please give an explanation.
Taxol was used as a standard control and Taxol is already commercially available; so, it would be sounding if we compare our optimized salt with a commercial product already in the market.
Web source/URL of docking software to be quoted as a reference.
There is no an available source/URL for the docking software.
Docking score of the pose selected need to be mentioned.
Docking score has now been provided.
Values highlighted in Figure 1 require explanation.
Figure 1 caption has now been modified accordingly
Reviewer 2 Report
The manuscript title "A Novel Curcumin Arginine Salt: A Solution for an Old Problem of Poor Solubility, and Potentiating Anticancer Activities" is well written and presented in an interesting manner. In the presented manuscript, docking studies, spectral and thermal analyses were used to characterize the curcumin arginine salts using different methods.
The paper will be improved by emphasizing the novelty in the Introduction section. What new question did this study set out to address and why is it important.
Author Response
We appreciate your constructive comments which are all valuable and very helpful for revising and improving our paper, as well as guiding our research. We have carefully considered comments and have revised the manuscript accordingly. The main revised portions of the manuscript are trackchanged, and our point-by-point responses to the comments are listed below.
The manuscript title "A Novel Curcumin Arginine Salt: A Solution for an Old Problem of Poor Solubility, and Potentiating Anticancer Activities" is well written and presented in an interesting manner. In the presented manuscript, docking studies, spectral and thermal analyses were used to characterize the curcumin arginine salts using different methods.
The paper will be improved by emphasizing the novelty in the Introduction section. What new question did this study set out to address and why is it important.
The novelty aspects have now been highlighted.
Reviewer 3 Report
Rewrite the abstract section, there are many spelling mistakes. The use of connectors is recommendable in order to present fluid ideas.
Line 23: slat/ion-pair?
Check the whole manuscript to avoid grammar mistakes. For example:
The authors repeat so many times the word “curcumin”.
Introduction: lines 46-48. Use “curcumin” 4 times in a paragraph of 4 lines. The same In the lines 50-52, and lines 59-63.
Line 124-125: It seems like the last sentence is related to the previous idea, so the point between “wettability” and “Due” is not correct.
Line 134: the degree symbol is not correct.
In some lines, the authors use the symbol “%” without space between it and the number, meanwhile, in other parts use space. The style must be homogenized.
Line 179: “Six consecutive concentrations (3.125 to 100 µg/mL) of curcumin, curcumin:L-argi- 179 nine Coppt and Taxol.” The idea seems to be incomplete.
The quality of the figures has to be improved.
The discussion of the results is very poor.
Author Response
We appreciate your constructive comments which are all valuable and very helpful for revising and improving our paper, as well as guiding our research. We have carefully considered comments and have revised the manuscript accordingly. The main revised portions of the manuscript are trackchanged, and our point-by-point responses to the comments are listed below.
Rewrite the abstract section, there are many spelling mistakes. The use of connectors is recommendable in order to present fluid ideas.
The abstract has now been corrected accordingly
Line 23: slat/ion-pair?
The phrase has been corrected.
Check the whole manuscript to avoid grammar mistakes. For example:
The authors repeat so many times the word “curcumin”.
Introduction: lines 46-48. Use “curcumin” 4 times in a paragraph of 4 lines. The same In the lines 50-52, and lines 59-63.
These sentences have now been revised and rewritten.
Line 124-125: It seems like the last sentence is related to the previous idea, so the point between “wettability” and “Due” is not correct.
The paragraph has now been corrected.
Line 134: the degree symbol is not correct.
The degree symbol has now been corrected.
In some lines, the authors use the symbol “%” without space between it and the number, meanwhile, in other parts use space. The style must be homogenized.
The style has been revised and now it is consistent throughout the manuscript.
Line 179: “Six consecutive concentrations (3.125 to 100 µg/mL) of curcumin, curcumin:L-argi- 179 nine Coppt and Taxol.” The idea seems to be incomplete.
The sentence has now been revised and corrected.
The quality of the figures has to be improved.
The resolution and quality of all figure have now been enhanced.
The discussion of the results is very poor.
The authors have been through the results and discussion section and it has now been improved.
Round 2
Reviewer 3 Report
The authors made the proper corrections. However, the discussion section is still very poor and could be enhanced by comparing the results obtained and those reported in the literature.
There are a few mistakes in the manuscript, for example:
Line 156: "Figure 5" in bold?
Line 164: "in vitro" must be in italic font.
Author Response
The authors would like to thank the reviewer for rigorous reviewing. Point-to-point responses are trackchanged in the text and repsonded as below:
The authors made the proper corrections. However, the discussion section is still very poor and could be enhanced by comparing the results obtained and those reported in the literature.
While we wish that the reviewer mentioned specific sections; the authors have been through the results and discussion section and several sections (docking, solubility and cytotoxicty) were revised and the results were compared with those mentioned in the literature.
There are a few mistakes in the manuscript, for example:
Line 156: "Figure 5" in bold?
The formatting has been corrected.
Line 164: "in vitro" must be in italic font.
in vitro has been italized throughout the manuscript